# Identifying Types of Dating Violence and Protective Factors among Adolescents in Spain: A Qualitative Analysis of Lights4Violence Materials

**DOI:** 10.3390/ijerph17072443

**Published:** 2020-04-03

**Authors:** Alfredo Pérez-Marco, Panmela Soares, Mari Carmen Davó-Blanes, Carmen Vives-Cases

**Affiliations:** 1PhD Candidate Faculty of Health Sciences, University of Alicante, 03690 Alicante, Spain; alfredoperezmarco@gmail.com; 2Department of Community Nursing, Preventive Medicine and Public Health and History of Science, University of Alicante, 03690 Alicante, Spain; mdavo@ua.es (M.C.D.-B.); carmen.vives@ua.es (C.V.-C.); 3Public Health Research Group, University of Alicante, 03690 Alicante, Spain; 4CIBER of Epidemiology and Public Health (CIBERESP), 28029 Madrid, Spain

**Keywords:** intimate partner violence, adolescent, qualitative analysis, protective factors

## Abstract

Introduction: The Lights4Violence project was created to promote healthy relationships among adolescents using a school intervention in which participants developed video-capsules where they highlighted skills to resolve situations of dating violence. This study aims to assess the results of the Lights4Violence training program by identifying different types of violence and positive development assets that Spanish adolescents use in their video-capsule scripts. Methods: A thematic analysis of the Lights4Violence video capsules was carried out. Open coding was used to identify violence patterns. A deductive analysis was used to identify student assets using the “Positive Youth Development Model”. Findings: Adolescents describe different patterns of violence, such as psychological violence, sexist violence or verbal violence that is present on the scripts. However, they showed themselves capable of resolving these situations using language and personal empowerment skills as resources. Family, friends and community were identified in adolescents’ scenarios as the most frequent assets to address situations of conflict. Conclusion: Adolescents can promote healthy relationships using protective factors against violence. Interventions that use this approach can potentially be useful in preventing violence.

## 1. Introduction

The Convention on Prevention and Combating Violence against Women and Domestic Violence of the Council of Europe defined Intimate Partner Violence as a pattern of aggressive and coercive behaviors, including physical, sexual and psychological acts, as well as economic coercion, which adults or adolescents use against their intimate partners without their consent [1]. In terms of the adolescent population, dating violence (DV) refers to Intimate Partner Violence between two people in a close relationship whose nature can be physical, emotional or sexual (including stalking). This includes situations between non-marital partners, including first dates and in heterosexual and same-sex relationships [2].

Statistics show that more than one in three women (37.3%) and men (30.9%) have experienced sexual violence, physical violence or stalking by an intimate partner in the United States between 2010 and 2012 [3]. Regarding the adolescent population, the number of high-school aged students that have been victims of physical or sexual DV has been decreasing during recent years, but there are significant differences between men and women, and between different ethnic groups [4]. At the European level one in five women has experienced the physical and sexual violence of a partner since the age of 15. Also, women between the ages of 18 and 29 have the highest prevalence of physical and/or sexual violence in the past 12 months. Psychological violence in a relationship affects up to 43% of women in the European Union (EU). Psychological violence is defined as controlling or abusive behavior, economic violence and blackmail with abuse of children. In that case, the age of the participants is not related to the answers received [5]. Spanish data have shown very similar results. Around 30% of adolescents, both boys and girls, are or have been involved in DV, although moderate forms of violence are more frequent than severe forms [6]. Psychological DV victimization is also most prevalent among young people (aged 14–20), with rates of up to 94% in girls and 93% in boys, while physical DV may affect up to 42% of girls and 39% of boys [7] 

There are several research studies regarding how DV affects the health outcomes of the adolescent population and its impact on adult lifestyles. Over the short term, victims and perpetrators of IPV tend to report lower rates of self-esteem and more feelings of self-blame, hurt, anxiety and anger. They also report communication problems, fewer problem-solving skills, and identify violence as a way to change the conduct of others [8]. All of these factors can result in the development of eating disorders, use of illicit substances and school failure [9]. Over the longer term, studies show that mental health problems including depressive mood and suicidal ideation, substance use, and IPV victimization in adulthood are longitudinal effects of DV during the adolescent ages [10,11]. These may occur in addition to the development of other psychopathological disorders such as paranoid ideation, psychosis or somatization [12]. Moreover, adolescents internalize disorders such as anxiety and depression during their dating relationships. They may also externalize their mental health problems to the detriment of the couple [13]. The World Health Organization (WHO) has prioritized mental health as a priority public health issue and established multi-sectoral plans to address it. These integral approaches include action on gender-related issues [14].

Two of the most recent meta-analyses about the effects of school interventions against DV in teenagers conclude that this type of intervention program is useful in order to gain knowledge about this issue in this population and improve the ability to identify it [15]. However, these prevention strategies have not been strong enough to decrease the incidence of DV. Thus, it is necessary to conduct studies to determine how these interventions affect adolescents’ skills to prevent this problem [16]. 

One such approach is the Positive Youth Development Model. This model is based on the idea that all teenagers have the potential for successful and healthy development and that human development is not predefined. There is always the possibility to change it [17]. The model not only defines the areas that result in healthy development (emotional, social, cognitive, moral and personal), but also the resources and experiences that promote it. These are defined as assets for development, and they can be related to one’s family, school, community or a person him/herself [18].

The European project “Lights, Camera and Action against DV” (Lights4Violence) began in this context. Its main objective is to promote adolescents’ ability to take part in healthier intimate relationships with their peers, prioritizing IPV-related protective factors that are present in adolescents themselves and in the context in which they have relationships (families, schools, and friendships). The project integrated an education-based intervention, *Filming Together to See Ourselves in a New Present*, which trained adolescents on competencies to establish healthy dating relationships and protective factors for youth development. The project included the creation of a final video-capsule in which adolescents described a violent situation in a dating relationship and included at least one asset for positive youth development and one or more personal competencies. These competencies were related to anti-sexism and violence rejection attitudes, problem-solving, empathy, communication skills and/or assertiveness that are useful for solving partner conflicts and that contribute to the development of healthier relationships [19].

The main objective of this article is to assess the results of the Lights4Violence training program by identifying different types of violence and positive development assets that Spanish adolescents use in their video-capsule scripts. This study will give us a first approximation of the results of the Lights4Violence training program among target adolescents in Spain. 

## 2. Materials and Methods 

This is a qualitative study based on an analysis of the scripts prepared by the students during the Lighs4Violence training program sessions. The program was carried out in five modules [19]. Each module contains between 15 and 17 sessions of approximately 50 min each. In the first two modules, we worked with students on different activities related to myths and social constructions of partner relationships, sexist attitudes, types of intimate partner violence, external protective assets (family, friends, teachers) and personal skills (empathy, assertiveness, problem-solving, and communication skills) of positive relationships. In the modules, students were given the knowledge and skills to write scripts and film the video-capsules. 

Analysis was carried out of the video capsule scripts that the students participating in the project developed for the activity Filming Together to See Ourselves in a New Present after the educational intervention. During the activity, student groups of four to five teenagers wrote a script for a video capsule. The script had to include a violent situation as well as an asset for healthy relationship development, which students had learned about during the theoretical sessions.

Using convenience sampling techniques, the students were recruited from a public high school in the northern zone of Alicante, Spain. The area is a vulnerable area of the city with high rates of school dropout, low family incomes, and high immigrant populations [20]. A total of 123 male (56%) and female students (44%) aged 13–15 participated. There were 85 students in their second year of compulsory secondary education (ESO) divided into 4 groups and 38 students in their third year of ESO divided into 2 groups. Around 18.8% of boys and 19% of girls reported DV experiences. 

A total of 15 scripts were developed. The scripts were transcribed from the original hand-written versions by the research team and analyzed using paper-based techniques. After a first reading to become familiar with the materials, double data coding and analysis was carried out. Analysis was carried out among peers to give more credibility to the identified categories. In the cases in which the peers disagreed on the results, a third researcher resolved discrepancies.

The identification of different types of violence required an open-coding and inductive process in which theory emerged from the analysis, and permitted identifying themes and sub-themes and establishing connections between them. Different definitions of violence were considered and matched with major themes. These are shown in Table 1 and were also included in the training program materials for students. 

A theoretical-deductive analysis was proposed to identify the assets students use to solve situations as a part of the Positive Youth Development Model proposed by Oliva et al. [23] and Scales and Leffert [18]. The categories established a priori for the training program materials and used later in this analysis can be found in Table 2. 

We decided to use thematic analysis due to the inductive nature of the technique and in order to establish and understand underlying attitudes and perceptions that can differ by culture [24]. The project was approved by the ethics committee of the University of Alicante.

## 3. Results

A total of 15 scripts were analyzed in which 12 represented heterosexual couples, 2 represented gay couples and one a lesbian couple. In 12 of the scripts the characters are unnamed. Different types of violence that occur in the adolescent population were identified as major themes, with associated conduct, attitudes, actions and representations. The Assets for Youth Development Model was used to identify different types of protective assets against violence related to adolescents’ relatives, communities and friends.

### 3.1. Psychological Violence is the Type of Violence Most Represented by the Adolescents

Psychological violence was the most represented type of violence by adolescents in the short film scripts, and situations of jealousy, blackmailing, ignoring a partner as punishment, lack of appreciation, non-acceptance and infidelity as revenge were quite common in the materials analyzed. These situations were represented as the causes for the violent situations or the result of it. Examples related to jealousy from the coded information included “he is feeling jealous because of the lack of control over her”, “she feels jealous because of misunderstanding a situation with him” or “he feels jealous because of the clothing she wants to wear”. These psychological violence patterns are in some of the cases not represented as isolated situations. A situation of jealousy could begin as a situation of blackmail in which one partner claims “that’s the way couples work”. Blackmailing situations also come from ignoring the partner as punishment, for example, “the boy leaves the situation angry threatening Reyad that things are not going to be like that”. In addition, the scripts show the use of infidelity as revenge for a past situation of violence, understanding it as not only a sexual pattern of violence, but also a psychological one, because the violent situation is not only the infidelity itself, it is related to a desired reaction from the other partner.

Another relevant type of violence is sexism It is represented as attitudes and speech and not always recognized as “sexist” by the adolescents. In only one of the scripts a neighbor appears who supports non-sexist behavior and equality. One of the most representative attitudes of sexism as violence is “control of what one wears”. Different scripts show couples in which a boy starts an argument because of the clothes a girl decides to wear. Moreover, there are also different sexist patterns demonstrated in the speech of the characters, like “a women is not going to touch me”, “You are a bitch […] I am going to tell everybody what kind of girl you are” or “I am your boyfriend and it is normal that you want to have sex with me”. 

Verbal violence in terms of “verbal discussions between couples” or “verbal discussions in groups” was used to describe violent situations. Literal examples were identified in all the scripts, represented as an action: “the discussion starts”, “they start talking in a loud voice”, “friends take part in the conversation and start telling them…”. Physical violence is only present in one of the stories, and it takes place between the parents of one of the characters. It can be understood that adolescents see this type of violence as part of older generations. Finally, sexual violence as sexual aggression is only represented in one of the stories and it is committed by a boy on a girl. 

Table 3 shows a synthesis of the qualitative thematic analysis carried out with the scripts.

### 3.2. Adolescents See Themselves as Able to Deal with Violent Situations

Concerning the Assets for Youth Development Model, Spanish adolescents seemed to see themselves as able to deal with many of the violent situations represented using different internal assets that they had been exposed to during the intervention.

Examples can be found in the recognition and management of their emotions. There are situations in the scripts in which adolescents recognise inappropriate attitudes and conduct and feel ashamed about them. “Rocío apologizes to Yerai and is ashamed of her behavior”, “she has never felt so bad for that kiss before”. On the other hand, adolescents show that they can identify problems with their partners and in terms of their own mistakes. They make decisions about it, for example, in the scripts: “the couple recognizes their mistakes and they realize they will not be able to continue their relationship because it will not be possible to forget what has happened” or “she finally sees the reality […] and decides she is not the kind of partner she wants to have”. 

Decision making is also an asset of the model. In the scripts, adolescents are represented as people who make decisions within their relationships with their peers. In their dating relationships this decision making involves establishing limits as a result of a discussion or conduct. Examples include, “I am fed up with your stories, I am going out and I am going to do whatever I want with or without you”, “the girl tells him she will give him a time to reflect”, or “she decides she is not the kind of partner she wants to have”. These situations are connected to feelings of autonomy and self-esteem, and many of them take place in women. Here it is possible to identify an empowered figure of a woman that can make decisions and communicate them in an assertive way using their self-esteem and autonomy. For example, “[…] she is not obliged to tell him everything she is doing, neither where she is nor who she is with”, and “I am fed up with your stories, I am going out, and I am going to do whatever I want with or without you”. There is also other speech related to autonomy and self-esteem in other kinds of characters, for example, in one of the stories a girl realizes that she must give more value to herself. In another story, a boy realizes he only sees his boyfriend as a friend and wants to leave the relationship in order to feel more autonomous.

Communication skills are also portrayed in the different stories. They are related to verbal discussions and are also related to assertiveness as most verbal communication-related skill and another relational skill on the conversations between the adolescents associated with non-verbal communication. In relation to verbal communication, many of the scripts contain explanations and clarifications related to the speech of the young people, for example, “Ampi tries to clarify what has happened, explaining to Rocio […]”, “the boyfriend explains that she was a friend […]”, etc. These clarifications are also given by the narrators: “Marcos is surprised by Cristina’s attitude and asks what’s wrong”. There is a nature of assertiveness of the young people, who state their opinions and thoughts with confidence: “[…] he tells him that he does not want a person like that in his life and he has to change his attitude”, “Listen to that, this has just finished because I am fed up with stories. I am going out, and I am going to do whatever I want with or without you”, “the boyfriend explains […] that this is not the way to act”. 

Adolescents themselves seem to be able to explain what is happening to third persons, as friends or community members. “She explains what has happened […]”. “The girlfriend feels blue and calls her friends.” Non-verbal communication skills are also represented in the scripts. One of the most common of the non-verbal skills among adolescents is moving aside when they start a verbal discussion. For example, in one of the scripts in which a family member is being physically abused, the boyfriend moves aside to let the mother and the daughter talk on their own. In one of the scenes it states, “Reyad is with their friends when his boyfriend appears. He moves over away for a moment from the group to go with him.” In one of the scripts, the way a girl establishes communication is written: “The girlfriend is always assertive (good communication, calm and self-confident)”. 

Finally, the scripts show a critical analysis of the situations by the adolescents which helps them to take the initiative and finally make decisions. So, for example, a girlfriend has a realization about the attitudes of her partner that help her make the decision to break up with him (“thanks to what happened, she saw a part of Marcos that she does not like”). A boy who was forcing his girlfriend to have sex with him realizes that she does not want to, and reflects (“finally the boy reflects and apologizes for his reaction”) to which she takes initiative and responds, “[…] she tells him seriously that she doesn’t want anything else at that moment”.

Regarding external assets, the family is the most present external asset in the different stories. The family is related to cases of infidelity and psychological violence towards the partner. Also, family members are shown that have a great sense of belonging to the characters (“[…] he doesn’t want to betray his sister”). This represents situations with positive communication and behavioral adjustment not only of the adolescents, but also towards other family members (“The daughter tells her mother that she cannot be like that”). Family members are shown to be a key to the autonomy, self-esteem and self-control of adolescents. Examples of this can be found in the stories where the family is present. For example, “siblings hug and he proposes to take a walk and have an ice cream,” and “sisters hug each other”, or “my sister is the most important thing for me”.

Other external assets related to the school and the community are also present in the stories. In situations of sexist violence, there is always a character that involves the community, such that the adolescent sees that sexist violence is a topic that affects everyone, sometimes represented by someone who is not related to the main characters. for example, in one of the stories a neighbor is the person who intervenes in a discussion, and another story involves a waitress who is also a psychology student. 

Regarding school assets, school counsellors are present in the stories. They are mediators of discussions. It is also interesting that teachers who have direct contact with the adolescents do not appear in the stories as school agents or facilitators. For example, in one of the stories a counsellor mediates between two adolescents, however the teacher is only shown as the person who sees them arguing without intervening between them.

Even though not considered on the model, it is important to mention that friends play different roles. On the one hand, in many violent situations, friends trigger discussions between partners. For example, “friends start discussing how she accepts his behavior. One of them tells her she saw him kissing another girl”. But, on the other hand friends act as mediators in conflicts (“their friends, witnesses of what was happening, approach and tell the boy that what he is doing is not right and has no justification”) or act as supports for the adolescents.

## 4. Discussion

The analysis of the scripts shows that adolescents in Spain represent different types of violence related to their dating relationships. In the scripts, it is possible to identify verbal violence, psychological violence, sexist violence, sexual violence and physical violence and related attitudes and behavior. Violent situations are usually resolved using the different assets presented by the Positive Youth Development Model, in which adolescents show themselves capable of dealing with obstacles or getting support from external agents to help them see their situations clearly. 

Regarding psychological violence, there is evidence that suggests that adolescents do not recognize this type as violence [25]. Studies report that 47% of adolescents consider that jealousy is a sign of love [26]. Regarding infidelity, adolescents reproduce patterns that occur in older generations [27]. They use infidelity as revenge to punish their partner for something they do not understand. This conduct has been labelled by Drigotas et al. as revenge-hostility infidelity [28]. Similar to what occurs with verbal violence, studies (in Spain) conclude that it is the pattern of violence most perceived by adolescents [29]. It is prevalent and normalized and not perceived as something alarming [30]. The “Síndic de Greuges” report states that family, teachers and students perceive that verbal violence is most frequent during the school ages [31].

The Spanish Centre of Sociological Research survey of 2012 related to gender violence perspectives among young people documents the social to intolerance sexist attitudes in Spain. The Spanish population identifies that women suffer from violence exclusively because they are women, and most survey respondents state that this is unacceptable [32]. We can also see these types of attitudes in the scripts analyzed. Sexist patterns that are present in the stories are usually identified, managed and resolved by the girls, who make the decision to empower themselves, which is at odds with the evidence that suggests adolescents do not recognize these violent attitudes and that they confuse being a victim of sexist violence with the idea of romantic love [33].

In relation to physical violence, in our scripts the students seem to feel that it is not a relevant problem for their generation. Seeing situations from an external point of view and with the idea of studying the problem, they identify resources and behavior patterns related to the appearance of that pattern [34], Statistics have shown that physical violence is less prevalent in the adolescent population [35] and that they identify physical violence as unacceptable in most cases [36]. In addition, intrafamily violence during childhood and adolescence is a predictive factor for the development of violence behavior in the future [37]. 

Our results also show that adolescents use internal assets to handle violent situations. They can manage their own situations and feelings and have skills for problem resolution after the intervention. This result is relevant because it does not correspond with the data found in the literature that suggests that adolescents who are victims or perpetrators of DV tend to report less self-esteem, reduced self-worth, increased self-blame, anger, hurt and anxiety, and ineffective communication and problem-solving skills [8,38,39,40,41]. It is also important to consider that the short film scripts were written in the context of an intervention designed to reinforce these types of competencies. 

Another important asset identified in the scripts is the family. Research suggests that Hispanic cultures give importance to the family and find relatives to be a protective factor against DV [42]. It is interesting that the situations in the scripts take place at school and include adults who take part in the violent situations. Teachers and counsellors appear less in the scripts. However, in most of the interventions carried out to prevent DV, teachers are the main agents of change and are in charge of education related to gender values. The support of teachers and of school staff during a violent situation can help determine how students feel about safety and their fears at school [43]. 

Finally, friends were represented in the scripts as the spark that sets off violent situations and as a main support for their peers in many difficult situations. Furthermore, group belonging is an important issue in the appearance and resolution of violence [44]. Friendship represents a key point in adolescent development, and they follow each other’s behavior [45].

Among the limitations of this study, it should be noted that the scripts of the short films were developed in the context of an intervention that required them to include protective factors. This could influence the consistency and credibility of the data. Nevertheless, students were free to mention different patterns of violence and protective factors that they considered relevant for their stories. Also, the results of this study may not be transferable as it was performed with a sample with two years of secondary education in a high school in Alicante. If the sample profile were to change (different years, educational institution, geographical area) we might obtain different results. Thus, more experiences of this type are needed to confirm the transferability of the results. 

## 5. Conclusions

After completing the Lights4Violence project intervention, students were capable of identifying different types of partner violence and protective assets needed to resolve them. Psychological violence was the most frequent type of violence represented by adolescents in their scripts. Furthermore, they saw themselves as primary agents in the resolution of violent situations, and they demonstrated the ability to identify external assets such as family, friends or school staff to deal with their problems. Adolescents also saw themselves as potential promoters of healthy relationships with their peers, using protective factors against violence as shown in the scripts of their video-capsules. Considering the importance of developing strategies to promote healthy partner relationships among adolescents, these firsts result of the Lights4Violence program show the potential of educational activities to give adolescents capabilities to identify and resolve situations of DV. An extension of the Lights4Violence study should include an analysis of the different materials obtained in the participant countries with a comparison of the results and the quantitative assessment of the pre-post effects on participants’ attitudes, personal and external assets to cope with partner violence. 

## Figures and Tables

**Table 1 ijerph-17-02443-t001:** Definitions of the different types of violence considered in this thematic analysis.

Verbal violence	A situation in which words and language are used with the intention to hurt someone or are likely to cause damage. Actions such as insults or verbal discussions are examples of this type of violence.
Physical violence	A situation in which physical force is used with the intention of hurting someone or to cause damage. Actions such as slapping, hitting, shoving or any other action that implies physical contact are examples.
Psychological violence	A situation in which violence is perpetrated with the intention of damaging to the dignity of a person. This can imply verbal and physical violence. In this case, actions such as coercion, blackmailing and ignorance are examples.
Sexist violence	A situation in which violence occurs because perpetrators and victims follow actions and attitudes that discriminate for reasons of gender.
Sexual violence	A situation of coaction or harassment of a person with the objective of having specific sexual conduct.

Note: adapted from Krug, Dahlberg, Mercy, Zwi and Lozano, 2003 [21]; Kilmartin and Allison, 2007 [22].

**Table 2 ijerph-17-02443-t002:** Categories for identification of assets that adolescents describe to resolve violent situations in their first sentimental relationships.

	**Internal Assets**
Personal	Assertiveness Relational skills Conflict resolution skills Communication skills Social commitment Responsibility Pro-social behavior Sense of justice Respect for diversity Self-esteem Self-concept Self-efficacy Self-control Autonomy Sense of belonging Personal initiative Empathy Recognition and management of emotions Frustration tolerance Optimism and sense of humor Capacity for critical analysis and analytical thinking Creativity Planning capacity Decision-making capacity
	**External Assets**
Family	Affection
	Adequate resolution of conflicts
	Establishment of boundaries
	Family support
	Positive communication
	Promotion of autonomy
School	Positive links with educators
	Affectionate and safe school climate
	Opportunities for participation and leadership in group activities
	Offer of programs that promote personal competences, social and emotional factors to face adolescence and contribute to integral development
Community	Safety
	Availability of structured extracurricular activities
	Positive assessment of the adolescent
	Assignment of responsibilities and roles to the adolescent
	Existence of resources and/or activities for young people

Note: adapted from Oliva et al., 2011 [23]; Scales and Leffert, 1999 [18].

**Table 3 ijerph-17-02443-t003:** Summary of the patterns of violence identified with their codes, categories and themes obtained after open-coding analysis.

Quotes Extracted from the Scripts of the Video Capsule	Codes	Categories	Themes
“They were arguing about why the boy was being unfaithful to take revenge on the girl, who had also been unfaithful to him a long time ago.” “Her boyfriend passes by and when he sees her there is a scene of jealousy.” “The boyfriend begins to blackmail her emotionally.” “The girlfriend arrives and tells her that she can’t meet for dinner because she was already there.” “[...] the girlfriend wants her to stay and he ignores her.”	He feels jealous because of lack of control over her. Emotional blackmailing as an idea of love. She feels jealous due to misunderstanding the situation with her partner. One of them prepares a plan and the other goes away. Lack of appreciation. Inequalities in the couple’s relationship. Lack of affection from the man towards the woman. The man does not accept that his partner wants to leave him. Emotional blackmail due to lack of control over the partner. Ignorance of the partner as a means of manipulation. Control of friendships due to jealousy. Infidelity as revenge for previous infidelities. She feels jealous of his behavior with a friend. Insecurity because of comments from a friend.	Jealousy Blackmailing Ignorance Lack of appreciation Non-acceptance of the situation Infidelity as revenge	Psychological violence
“[…] screaming, he tells her that ‘because he is very light in clothes’.” “[…] he threatens to tell everyone the type of girl she is.” “[...] a girl is not going to touch me.”	Control of clothing with sexist arguments. Sexist attitude as a control argument. Discussion with threats and sexist arguments.	Comments related to patriarchy Clothing control	Sexist violence
“The boy, ignoring what she is telling him, keeps kissing and touching her.”	A boy is unfaithful to his partner with her sister. Sexual situations in which he wants to continue with something, and she does not	Sexual assault Infidelity itself	Sexual violence
“The other boy does not understand and begins to argue.” “The boy leaves angry threatening Reyad that ‘things will not stay like this’.” “[…] Three friends position themselves, and three support the boyfriend who is also there. (A discussion begins about the topic)”	Verbal discussion. Discussion with threats Discussion with sexist arguments. Verbal discussion in groups.	Discussion between peers Group discussion	Verbal violence
“When she was going to throw on the girl…” “Marcos reacts fatally and catches him by the neck of the shirt.” “[...] the father comes out like a bad beast slamming the door. When they enter, they see the mother crying with signs of physical aggression.”	Physical violence towards the woman for possible infidelity. Physical violence between parents that leads to not accepting a relationship.	Physical aggression between peers Physical aggression between parents	Physical violence

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
