# Peer review of "Identifying Types of Dating Violence and Protective Factors among Adolescents in Spain: A Qualitative Analysis of Lights4Violence Materials"

_ijerph, 2020, doi:10.3390/ijerph17072443_

Round 1

Reviewer 1 Report

Globally I consider the article is interesting, as it reports a very specific kind of methodology and how it can be used to inform future interventions. It is very well written, as the contents are very well integrated, it is an article very ease to read.

I have two main aspects I would like the authors to consider related with the objectives of the study and how they are related with the conclusions: 1) the objective of the study is stated is lines 86-87, but in the conclusions authors refer several times to the ability of the adolescents to maje use of the assets in the videoclips; so my question is, why did the authors not stated as another objective of the study not only the characterization of the types of violence and assets present, but also to evaluate the impact of the training in the final videoclips? I question this because the authors refer that the training included information about specific competences and assets, and the objective of the training was also to show it in the final videos, right? In this perspective I think it would be clear and more consistent with the conclusions to clarify if the objective was also to evaluate the impact of the training; the final recommendations of the use of this methodology with adolescents would be more consistent

2) I think the sample of adolescents is not well characterized; it is necessary to let the readers know who were these students, in order to legitimate some specific aspects of the conclusions as I detail below.

Besides these general points, I detail some other specific aspects that I think would improve the article:

Pg2, line 47: Given that the authors highlight in the title of the article that the study focuses on Spanish adolescents, it would be pertinent for this contextual framework to also refer to the current reality in Spain in order to make clear the relevance of the work.

Please review the references list as it has some minor incorrections

Pg2, line 53: these disorders are those reported in the cited reference, mas they are not the only ones... I suggest you can insert "Among other psychopathological disorders, such as ..." (and refer other recent references such  as Dacila, Capaldi & Greca (2016), or Clements, Bennett & Hungerford (2019).

Pg 2, line 56: A different citation format?

Pg 2, line 58: sentence incomplete?

Pg 2, line 91: uncorrect writing in the name of the project

Pg 3, line 99: I think it would be very important to characterize the participants; age, year of schooling attended, if some had already been flagged for dating violence, presence of psychopathology; it would also be important to mention with which criteria were they selected

Pg 7, line 189: caps not necessary in "To"

Pg 8, line 257: to is not in the correct place

Pg 9, line 258: it lack the "are" (because they are women?)

Pg 9, lines 273-276: I think this can be stated only if authors caractherize specifically the adolescents that participated in the study in terms of being victims or perpetrators od dating violence before, and this is not present in the article.

Author Response

#1 Comments and Suggestions for Authors

Globally I consider the article is interesting, as it reports a very specific kind of methodology and how it can be used to inform future interventions. It is very well written, as the contents are very well integrated, it is an article very ease to read.

I have two main aspects I would like the authors to consider related with the objectives of the study and how they are related with the conclusions:

1) the objective of the study is stated is lines 86-87, but in the conclusions authors refer several times to the ability of the adolescents to maje use of the assets in the videoclips; so my question is, why did the authors not stated as another objective of the study not only the characterization of the types of violence and assets present, but also to evaluate the impact of the training in the final videoclips? I question this because the authors refer that the training included information about specific competences and assets, and the objective of the training was also to show it in the final videos, right? In this perspective I think it would be clear and more consistent with the conclusions to clarify if the objective was also to evaluate the impact of the training; the final recommendations of the use of this methodology with adolescents would be more consistent.

Thank you for your comment. We agree that video-capsules scripts content which was analyzed in this study is a fruitful way to assess the results of our training program in Lights4Violence project. Consequently, we reformulated the objective as following: “The main objective of this article is to assess the results of the Lights4Violence training program by identifying different types of violence and positive development assets that Spanish adolescents use in their video-capsule scripts” (see abstract and “Introduction” page 2).

2) I think the sample of adolescents is not well characterized; it is necessary to let the readers know who were these students, in order to legitimate some specific aspects of the conclusions as I detail below.

Ok. We added the following information about participants: “Using convenience sampling techniques, the students were recruited from a public high school in the northern zone of Alicante, Spain. The area is a vulnerable area of the city with high rates of school dropout, low family incomes, and high immigrant populations [20]. A total of 123 male (56%) and female students (44%) ages 13-15 participated. There were 85 students in their second year of compulsory secondary education (ESO) divided into 4 groups and 38 students in their third year of ESO divided into 2 groups. Around 18.8 percent of boys and 19 percent of girls reported that they had ever been exposed to dating violence experiences.”. (See “Methods”, page 4, lines 109-115).

Besides these general points, I detail some other specific aspects that I think would improve the article:

Pg2, line 47: Given that the authors highlight in the title of the article that the study focuses on Spanish adolescents, it would be pertinent for this contextual framework to also refer to the current reality in Spain in order to make clear the relevance of the work.

Ok. We added this information at the end of the paragraph (See page 2, lines 47-52)

Please review the references list as it has some minor incorrections

Pg2, line 53: these disorders are those reported in the cited reference, mas they are not the only ones... I suggest you can insert "Among other psychopathological disorders, such as ..." (and refer other recent references such  as Dacila, Capaldi & Greca (2016), or Clements, Bennett & Hungerford (2019).

Thank you for your suggestion. We added the reference and its content as you had suggested in the new version of the manuscript (See page 2, lines 61-64).

Pg 2, line 56: A different citation format?

Thank you for this correction. We changed it.

Pg 2, line 58: sentence incomplete?

Thank you for this correction. We changed it.

Pg 2, line 91: uncorrect writing in the name of the project

Thank you for this correction. We changed it.

Pg 3, line 99: I think it would be very important to characterize the participants; age, year of schooling attended, if some had already been flagged for dating violence, presence of psychopathology; it would also be important to mention with which criteria were they selected

We added this information in the second paragraph of methods section (see page 2, lines 111-115). We didn’t collect information about psychopathologies and these were not an inclusion criteria for participating in our program. 

Pg 7, line 189: caps not necessary in "To"

Thank you for this correction. We changed it.

Pg 8, line 257: to is not in the correct place

Thank you for this correction. We changed it.

Pg 9, line 258: it lack the "are" (because they are women?)

Thank you for this correction. We changed it.

Pg 9, lines 273-276: I think this can be stated only if authors caractherize specifically the adolescents that participated in the study in terms of being victims or perpetrators od dating violence before, and this is not present in the article.

We agree and we reformulated the conclusion as it follows: “After completing the Lights4Violence project intervention, students were capable of identifying different types of partner violence and protective assets needed to resolve them. Psychological violence was the most frequent type of violence represented by adolescents in their scripts. Furthermore, they saw themselves as primary agents in the resolution of violent situations, and they demonstrated the ability to identify external assets such as family, friends or school staff to deal with their problems. Adolescents also saw themselves as potential promoters of healthy relationships with their peers, using protective factors against violence as shown in the scripts of their video-capsules. Considering the importance of developing strategies to promote healthy partner relationships among adolescents, these firsts result of the Lights4Violence program show the potential of educational activities to give adolescents capabilities to identify and resolve situations of dating violence. An extension of the Lights4Violence study should include an analysis of the different materials obtained in the participant countries with comparison of the results. “(See Conclusions on page 9).

Reviewer 2 Report

The work presented, does not have the level and quality necessary to be published in this magazine. It should be improved with respect to the rationale and method, the qualitative analysis is not carried out I consider it adequate and the methodology is not described correctly.

Methods: A Thematic analysis of the Lights4Violence video capsules was carried out. Open coding was used to identify violence patterns. A deductive analysis was used to identify student assets using the “Positive Youth Development Model”. 

A work with this methodology does not contribute absolutely anything. It is the previous step of a scientific investigation. I do not believe that the analysis of the sequences of an intervention program is the subject of a scientific publication of this level.

The authors should test the validity of the tool with a qualitative or quantitative study.

The authors should do a qualitative study based on the scientific method.

The authors provide something about the validity of the program.

It is not a work with the scientific quality of the magazine.

Proper justification not provided, revision is outdated and limited. It does not include an informed theoretical model.

It does not indicate the number or who performs the analysis of the program.

They cannot pretend to make an investigation phase a result. That is part of the investigation procedure, not the result.

I urge authors to apply the program and to propose a research design to do research with specific results in the application of the program.

Author Response

#2 Comments and Suggestions for Authors

The work presented, does not have the level and quality necessary to be published in this magazine. It should be improved with respect to the rationale and method, the qualitative analysis is not carried out I consider it adequate and the methodology is not described correctly.

Methods: A Thematic analysis of the Lights4Violence video capsules was carried out. Open coding was used to identify violence patterns. A deductive analysis was used to identify student assets using the “Positive Youth Development Model”. A work with this methodology does not contribute absolutely anything. It is the previous step of a scientific investigation. I do not believe that the analysis of the sequences of an intervention program is the subject of a scientific publication of this level. The authors should test the validity of the tool with a qualitative or quantitative study. The authors should do a qualitative study based on the scientific method. The authors provide something about the validity of the program. It is not a work with the scientific quality of the magazine. Proper justification not provided, revision is outdated and limited. It does not include an informed theoretical model. It does not indicate the number or who performs the analysis of the program. They cannot pretend to make an investigation phase a result. That is part of the investigation procedure, not the result. I urge authors to apply the program and to propose a research design to do research with specific results in the application of the program.

Thank you for your comments. The main objective of this article is to use a qualitative approach to analyze the impact of Lights4Violence training program through identifying different types of violence and positive development assets that Spanish adolescents use in their video-capsules scripts. We considered that the used qualitative approach gives us an enrich information about the impact of our training program. Of course, the quantitative evaluation of the impact of Lights4Violence in our participant’s attitudes and personal skills to solve conflicts among partners and develop healthier relationships is relevant, but it will do in another study.

Reviewer 3 Report

Dear authors,

This paper represents an interesting qualitative study about the knowledge and strategies adolescents use to cope with dating violence. It takes part of a dating violence prevention program developed in Spain. To date, the number of intervention programs carried out in the country is scarce, so this study contributes to filling this gap in the research.

In overall, the study is well organized, although minor errors have been identified. I suggest authors check the paper, particularly pages two (line 46, line 56) and three (line 112).

I address some recommendations for authors in order to improve the quality of the study: 

  1.  The aim of the study. The authors state that the objective of the study is "to analyze different types of violence and positive development assets of that Spanish adolescents use in the short film of the Lights4Violence Project". The authors indeed describe the types of violence students identify, but it is crucial to underline that they do these video-capsules after they receive the intervention. My question is: If this activity ( the scripts and the video-capsules) is developed after the intervention. Should it be considered as a part of the project assessment? Do the participants identify these forms or types of violence because they participated in the intervention program? Do the researchers hypothesize that students would report these types of violence if they would not be part of the project? I think authors need to redefine the aim of the study, considering this study as a first evaluation of the program. By doing this, the discussion could enrich by including strengths and limitation of this first qualitative evaluation of the program and the need to carry out different procedures to assess the efficacy of the intervention programs. 
  2. Sample: authors need to describe in-depth the sample characteristics: number of participants, age, gender, etc. Besides, some questions need to be answered. If these students are part of the program, How many students participate in the scripts and video capsules, the total sample or just a subsample of participants? If they are a subsample: Did the researchers follow any procedure to select these students or were they volunteers? If they are a subsample of the participants. Were they different from the total sample in terms of age, gender, romantic experience, previous involvement in dating violence, for example?  
  3. Procedure: authors need to describe the program. How many lessons did the students receive? What were the main contents the program address? Did the students receive any instruction and support to carry out the video-capsules? Did the students develop the video-capsule just after the intervention or some weeks later? 
  4. Research design: the authors state that they use a quasi-experimental design, but they do not describe if they have experimental and control groups.  
  5. Results: Because this is a qualitative and content-based analysis, authors must present the Kappa index and the procedure they followed to resolve disagreements. Moreover, to understand the results better, authors must describe if the types and categories of violence they identified were addressed during the intervention program. The same can be said for the assets. Does the program content include all the assets? 
  6.  Conclusions: As said previously, the authors can enrich this section if they describe better the context of the intervention. As it is in the original version, the discussion does not go further in the field of dating violence. What is the main contribution of the study? I suggest a revision of this section. 

Author Response

#3 Comments and Suggestions for Authors

This paper represents an interesting qualitative study about the knowledge and strategies adolescents use to cope with dating violence. It takes part of a dating violence prevention program developed in Spain. To date, the number of intervention programs carried out in the country is scarce, so this study contributes to filling this gap in the research.

In overall, the study is well organized, although minor errors have been identified. I suggest authors check the paper, particularly pages two (line 46, line 56) and three (line 112).

Thank you. We revised both pages.

I address some recommendations for authors in order to improve the quality of the study:

 The aim of the study. The authors state that the objective of the study is "to analyze different types of violence and positive development assets of that Spanish adolescents use in the short film of the Lights4Violence Project". The authors indeed describe the types of violence students identify, but it is crucial to underline that they do these video-capsules after they receive the intervention. My question is: If this activity ( the scripts and the video-capsules) is developed after the intervention. Should it be considered as a part of the project assessment? Do the participants identify these forms or types of violence because they participated in the intervention program? Do the researchers hypothesize that students would report these types of violence if they would not be part of the project? I think authors need to redefine the aim of the study, considering this study as a first evaluation of the program. By doing this, the discussion could enrich by including strengths and limitation of this first qualitative evaluation of the program and the need to carry out different procedures to assess the efficacy of the intervention programs.

Thank you very much for your suggestion. We reformulated the study aim as it follows: “The main objective of this article is to assess the results of the Lights4Violence training program by identifying different types of violence and positive development assets that Spanish adolescents use in their video-capsule scripts” (see abstract and “Introduction” page 2).

Sample: authors need to describe in-depth the sample characteristics: number of participants, age, gender, etc. Besides, some questions need to be answered. If these students are part of the program, How many students participate in the scripts and video capsules, the total sample or just a subsample of participants? If they are a subsample: Did the researchers follow any procedure to select these students or were they volunteers? If they are a subsample of the participants. Were they different from the total sample in terms of age, gender, romantic experience, previous involvement in dating violence, for example? 

We added this information in the second paragraph of methods section (see page 3 lines 109-115).

Procedure: authors need to describe the program. How many lessons did the students receive? What were the main contents the program address? Did the students receive any instruction and support to carry out the video-capsules? Did the students develop the video-capsule just after the intervention or some weeks later?

This information was previously described in the study protocol elsewhere (see ref 19). However, we also added a summary on the current version of the paper as it follows: “Each module contains between 15 and 17 sessions of approximately 50 minutes each. In the first two modules, we worked with students on different activities related to myths and social constructions of partner relationships, sexist attitudes, types of intimate partner violence, external protective assets (family, friends, teachers) and personal skills (empathy, assertiveness, problem-solving, and communication skills) of positive relationships. In the modules, students were given the knowledge and skills to write scripts and film the video-capsules” (see methods page 3 lines 98-103).

Research design: the authors state that they use a quasi-experimental design, but they do not describe if they have experimental and control groups. 

This information was previously described in the study protocol elsewhere (see ref 19). However, we didn’t include it in this paper because it was not related with our study aim and we considered that it may be confusing.

Results: Because this is a qualitative and content-based analysis, authors must present the Kappa index and the procedure they followed to resolve disagreements.

We didn’t calculate a Kappa index. As it is explained in the current version of the manuscript, two of us did the analyses in parallel and a third researcher was consulted in case of disagreements (see page 3, lines 117-120).

Moreover, to understand the results better, authors must describe if the types and categories of violence they identified were addressed during the intervention program. The same can be said for the assets. Does the program content include all the assets?

Thank you for your suggestion. We added this information in the current version of the paper (see Methods pages 3 lines 124-125 and 4 lines 131-132).

 Conclusions: As said previously, the authors can enrich this section if they describe better the context of the intervention. As it is in the original version, the discussion does not go further in the field of dating violence. What is the main contribution of the study? I suggest a revision of this section.

Thank you for your comment. We reviewed the conclusion and re-write it as it follows: “After completing the Lights4Violence project intervention, students were capable of identifying different types of partner violence and protective assets needed to resolve them. Psychological violence was the most frequent type of violence represented by adolescents in their scripts. Furthermore, they saw themselves as primary agents in the resolution of violent situations, and they demonstrated the ability to identify external assets such as family, friends or school staff to deal with their problems. Adolescents also saw themselves as potential promoters of healthy relationships with their peers, using protective factors against violence as shown in the scripts of their video-capsules. Considering the importance of developing strategies to promote healthy partner relationships among adolescents, these firsts result of the Lights4Violence program show the potential of educational activities to give adolescents capabilities to identify and resolve situations of dating violence. An extension of the Lights4Violence study should include an analysis of the different materials obtained in the participant countries with comparison of the results“ (See Conclusions on page 9).

Round 2

Reviewer 2 Report

I consider that this part of the study is not sufficient to be eligible for publication in a journal of this level. As the authors point out, I hope that they send the truly interesting of their work. The application of the instrument to see the benefits. What they show in this work are research processes, not scientific information and results of interest for practice.

Author Response

#2 Comments and Suggestions for Authors

Point 1: I consider that this part of the study is not sufficient to be eligible for publication in a journal of this level. As the authors point out, I hope that they send the truly interesting of their work. The application of the instrument to see the benefits. What they show in this work are research processes, not scientific information and results of interest for practice.

Response 1: Our results present relevant information about an educational intervention that can be used as a model in other studies.  As we explained in the revised version of the manuscript, this is a first approximation that let us to assess the results of the Lights4Violence training program on target adolescents (see introduction page 2 lines 94-95). The main aim of this paper is not to quantify the impact of the training program on the students, but, to understand how they integrate and underlie program contents on their reality within the production of their own materials. It means, how they perceive violence and the resolution of the problems after an educational intervention. The use of qualitative thematic analysis is explained on page 4, lines 136-138. However, we recognize the limits of this approximation and we included them in the discussion section (see page 10 lines 313-320 and 334-335). 
